# Mitochondrial Biogenesis and Mitochondrial Reactive Oxygen Species (ROS): A Complex Relationship Regulated by the cAMP/PKA Signaling Pathway

**DOI:** 10.3390/cells8040287

**Published:** 2019-03-27

**Authors:** Cyrielle Bouchez, Anne Devin

**Affiliations:** 1Université Bordeaux, IBGC, UMR 5095, 33077 Bordeaux cedex, France; cyrielle.bouchez@ibgc.cnrs.fr; 2Institut de Biochimie et Génétique Cellulaires, CNRS UMR 5095, 1, rue Camille Saint Saëns, 33077 Bordeaux Cedex, France

**Keywords:** mitochondrial biogenesis, ROS, cAMP signaling, yeast

## Abstract

Mitochondrial biogenesis is a complex process. It requires the contribution of both the nuclear and the mitochondrial genomes and therefore cross talk between the nucleus and mitochondria. Cellular energy demand can vary by great length and it is now well known that one way to adjust adenosine triphosphate (ATP) synthesis to energy demand is through modulation of mitochondrial content in eukaryotes. The knowledge of actors and signals regulating mitochondrial biogenesis is thus of high importance. Here, we review the regulation of mitochondrial biogenesis both in yeast and in mammalian cells through mitochondrial reactive oxygen species.

## 1. Introduction

### 1.1. Mitochondrial Oxidative Phosphorylation

In eukaryotic cells, energy conversion processes are mandatory for both cell biomass generation and cellular maintenance. Adenosine triphosphate (ATP) is the cellular energy currency, and mitochondria play a crucial role in ATP synthesis thanks to the oxidative phosphorylation system (OXPHOS) located in the mitochondrial inner membrane. The oxidative part of this energy conversion process takes place in the four enzymatic complexes of the mitochondrial respiratory chain and leads to substrates—nicotinamide adenine dinucleotide (NADH) and flavin adenine dinucleotide (FADH2)—oxidation. Activity of the respiratory chain is based on the transfer of two electrons from above-mentioned substrates to their final acceptor, the dioxygen. These redox reactions are coupled to proton extrusion at the level of complexes I, III, and IV in mammalian cells (Figure 1A); thus, thanks to the inner mitochondrial membrane being quite impermeable, a proton gradient is generated across this membrane. The yeast *Saccharomyces cerevisiae* mitochondria do not harbor a proton-pumping complex I but rather a number of dehydrogenases (Figure 1B) in the inner mitochondrial membrane [1]. The phosphorylating part of this process involves the ATPsynthase, the adenine nucleotide translocator (ANT), and the phosphate carrier. The proton gradient generated by the mitochondrial respiratory chain activity is used by the ATPsynthase for ATP synthesis from ADP and Pi and by a number of mitochondrial carriers.

### 1.2. Mitochondrial ROS Production

The main superoxide (O_2_·^−^) producer in the cell is the mitochondrial respiratory chain. This production leads to hydrogen peroxide (H_2_O_2_) and hydroxyl radical (·HO) formation. Superoxide formation occurs when a unique electron is accepted by the ground state oxygen. Consequently, any electron transfer that involves a unique electron is susceptible of superoxide generation. This is particularly true in membranes, where oxygen solubility is high. The respiratory chain complexes I and III have long been studied for their contribution to mitochondrial reactive oxygen species (ROS) production [3,4,5,6,7,8]. In complex III, superoxide formation is due to the Q cycle [9], and ROS production can take place both in the mitochondrial matrix and in the intermembrane space. Besides the contribution of these complexes to mitochondrial ROS production, mitochondrial dehydrogenases are involved in ROS formation as well. For example, α-ketoglutarate dehydrogenase catalyses the oxidation of α-ketoglutarate in succinyl-coA with the generation of NADH. Single electron transfer occurs within this enzyme, and its activity can thus be associated with ROS production in the Krebs cycle [10]. For the same reason, the mitochondrial glycerol-3-phosphate dehydrogenase (G3PDH) produces ROS [11,12]. As stated above, the yeast *Saccharomyces cerevisiae* harbors two NADH-dehydrogenases, which are sites of ROS production as well.

In this review, we will address the relationships between mitochondrial ROS production and the regulation of mitochondrial biogenesis (i.e., the biogenesis of the whole compartment and more particularly of the OXPHOS system) both in yeast and in mammalian cells.

## 2. Mitochondrial Compartment Biogenesis

Mitochondrial biogenesis is a complex process. Indeed, mitochondria are organelles that harbor their own genome (mtDNA). In mammalian cells, mtDNA is a circular molecule, which encodes for 13 mRNAs, 22 tRNAs, and 2 rRNAs. All 13 mRNAs of mtDNA encode 11 subunits of the ETC complexes I (7), III (1) and IV (3), and 2 subunits of ATP synthase (complex V). However, mitochondria are genetically semiautonomous and strongly rely on the nuclear genome for their biological function. Thus, mitochondrial biogenesis necessitates the coordinated expression of both mitochondrial and nuclear genomes.

In the yeast *Saccharomyces cerevisiae* (Figure 2), mitochondrial biogenesis is regulated at the transcriptional level, by nuclear proteins. The main actor of this process is a transcription complex composed of four proteins that belong to the same family: heme activator proteins (Hap) 2, 3, 4, and 5 [13,14,15,16,17]. Hap2p, Hap3p, and Hap5p form a complex that is bound to nuclear DNA, and Hap4p is the co-activator of this complex, a functional homolog of peroxisome proliferator-activated receptor γ (PPARγ) coactivator-1 (PGC-1α, see below). Hence, activity of the overall complex is mostly dependent on Hap4p. The HAP complex regulates the expression of genes encoding several proteins such as proteins of the Krebs cycle or proteins of the OXPHOS system [18]. Mitochondrial genes expression depends on the RNA polymerase Rpo41 and its accessory transcription factor Mtf1 [19]. mRNA translation is regulated by specific translation factors such as Pet111 for COX2, Pet309 or Mss51 for COX1, and Atp22 for ATP6/8 [20]. These factors regulate protein synthesis, assembly, and function [21,22]. The homologue of the well-known mammalian transcription factor, the mitochondrial transcription factor A (TFAM), does not interact with the transcriptional machinery but seems to be implicated in mtDNA replication [23].

In mammalian cells (Figure 3), the main actors of mitochondrial biogenesis are the Nuclear Respiratory Factors (NRF), in particular NRF1 [25,26]. The NRF transcription factors are known to be involved in the transcription of several mitochondrial genes in particular genes encoding subunits of the mitochondrial respiratory chain complexes. Moreover, mitochondrial biogenesis is also regulated by the transcriptional family of peroxisome proliferator-activated receptor γ (PPARγ) coactivator-1 (PGC-1). This family is composed of PGC-1α, PGC-1-related coactivators (PRC), and PGC-1β. These proteins can interact with other transcription factors involved in the expression of mitochondrial proteins. Moreover, PGC-1 transcription factors activators interact with transcription factors such as NRF1 and 2 and regulate the expression of these proteins. These nuclear transcription factors regulate the transcription of the majority of mitochondrial proteins, including proteins that are required for the transcription of the mitochondrial genome such as TFAM (or mtTFA). The mtDNA encodes 13 components of OXPHOS system complexes: ND1, ND2, ND3, ND4, ND4L, ND5, ND6 (subunits of complex I); CYTb (subunit of complex III); COX1, COX2, COX3 (subunits of complex IV); ATP6, ATP8 (subunits of ATPsynthase) [27]; as well as 22 tRNA and 2rRNA [28].

## 3. ROS-Induced Downregulation of Mitochondrial Biogenesis in Yeast

### 3.1. cAMP Signaling and Mitochondrial Biogenesis

In the yeast *Saccharomyces cerevisiae*, the cyclic adenosine monophosphate (cAMP) signaling pathway can be activated by two distinct mechanisms. The first one is an extracellular glucose dependent system based on G-protein-coupled glucose receptor (Gpr1) activity. This G-protein-coupled receptor (GPCR) system activates adenylate cyclase, i.e., the enzyme that catalyses cAMP synthesis, which leads to an increase in intracellular cAMP [29]. The second one depends on the activity of the small RAS proteins regardless of the carbon source present in the medium [29]. These proteins belong to the small guanosine triphosphatases (GTPases) family, therefore their activity depends on GTP/GDP exchange. Yeast exhibits two Ras isoforms: Ras1 and Ras2, highly homologous to mammalian RAS. The adenylate cyclase is activated by RAS-GTP. Yeast protein kinase A (PKA) is a hetero-tetramer composed of two regulatory subunits and two catalytic subunits. The catalytic subunits are encoded by three genes: Tpk1, Tpk2, and Tpk3 [30], and the regulatory subunit is encoded by the Bcy1 (bypass of cAMP requirement) gene [31] (Figure 4).

Previous work from our laboratory has shown a tight link between the activity of this signaling pathway and the cellular content in mitochondria [24,32,33,34,35]. It should be stressed here that throughout this review, mitochondrial content will refer to the cellular amount of mitochondrial cytochromes, a robust parameter that allows cellular quantitation of mitochondrial respiratory chain amount [36]. Whenever such a quantitation is not doable, more often than not by restriction on available “material”, cellular mitochondrial content is assessed by citrate synthase activity measurement, a well-recognized method for such a quantitation [37,38,39]. The relationship between the activity of this signaling pathway and the cellular content in mitochondria is such that mutants exhibiting an increase in this pathways’ activity have an increased amount of mitochondria, whereas mutants exhibiting a decrease in this pathways’ activity have a decreased amount of mitochondria. Indeed, Dejean et al. showed that a disruption of the small GTPases or their inhibitors, IRA1 and IRA2 (Inhibitory Regulator of the RAS-cAMP pathway 1 and 2), triggers a reduction in cellular mitochondrial content, whereas an over-activation of these GTPases increases this content [40]. Moreover, in an OL556 strain, which is mutated in the *cdc25 gene* and the *rca1* allele of the PDE2 gene [41,42], allowing modulation of cAMP intracellular concentration by addition in the extracellular medium, the increase in intracellular cAMP concentration increases both the growth and respiration rate, in non-fermentable medium [43]. In such a medium, growth is directly dependent on ATP production by the OXPHOS [36]. Thus, an increase in both growth and respiratory rate points to an increase in mitochondrial OXPHOS activity. Moreover, we have shown that an increase in intracellular cAMP concentration induces an increase in both mitochondrial enzyme activities and mitochondrial cytochromes. This raised the question of the origin of the increase (or decrease) in cellular mitochondrial content upon cAMP signaling. Stationary state mitochondrial content is the result of two processes: its biogenesis and its degradation. Since no mitophagy (degradation) could be assessed, we assessed the regulation of mitochondrial biogenesis (activity of the HAP complex) when cAMP intracellular concentration was modulated (OL556 strain, see above). Exogenous cAMP addition to this strain increases HAP complex activity and thus mitochondrial biogenesis. cAMP triggers an enhancement of Hap4p stability, the regulator of the HAP complex [32]. Consequently, an increase in the cAMP pathway activity positively regulates mitochondrial biogenesis.

### 3.2. cAMP Signaling and Oxidative Stress

As stipulated above, previous work from our laboratory has shown a tight link between the activity of the cAMP signaling pathway and the cellular content in mitochondria. This was done through modulation of the activity of this pathway upstream of the PKA. Yeast has three A kinase catalytic subunits, which have greater than 75% identity and are encoded by the TPK (TPK1, TPK2, and TPK3) genes [30]. Although they are redundant for viability and functions such as glycogen storage regulation, the three A kinases are not redundant for other functions [44,45,46,47]. In the absence of the yeast protein kinase Tpk3p, there is a significant decrease in cellular mitochondrial content, when cells are grown in non-fermentable medium whereas deletion of either Tpk1 or Tpk2 has no consequences on mitochondrial activity/amount [47]. This points to a specific function of Tpk3p in the regulation of cellular mitochondrial content. Further studies allowed us to show that the downregulation of mitochondrial content in the absence of Tpk3p is due to an oxidative stress that originates at the mitochondrial level [47]. Moreover, reconstitution of cAMP-induced signaling on isolated mitochondria allowed us to show that Tpk3p-induced phosphorylation was involved in the regulation of mitochondrial ROS production i.e., in the absence of Tpk3p, mitochondrial ROS were increased [33]. Thus, an alteration of the cAMP pathway through *Tpk3* deletion induces a mitochondrial oxidative stress. It should be stressed that in this model, the priming event is the increase in mitochondrial ROS due to a defect in Tpk3p-induced phosphorylation at the level of mitochondrial respiratory chain. This increase then leads to a decrease in mitochondrial biogenesis.

### 3.3. Oxidative Stress and Mitochondrial Biogenesis Regulation

In order to decipher the molecular mechanisms involved in the decrease of cellular mitochondria content upon oxidative stress in the ∆tpk3 strain, regulation of mitochondrial biogenesis in this strain was studied. Enhancement of ROS production in this strain triggers a decrease in the HAP complex activity that correlates with a decrease in the cellular amount of the Hap4p protein, the co-activator of the HAP complex (Figure 5). Evidence that the decrease in mitochondrial biogenesis originates in Hap4p decrease came from a full reversion of this decrease upon overexpression of this protein [33]. Further, it was shown that this protein’s stability is highly decreased upon any kind of oxidative stress applied to cells [34]. Evidence that ∆tpk3-induced mitochondrial ROS were responsible for the decrease came from the restoration of a wild type phenotype (activity of the complex, stability of Hap4p, mitochondrial content) in the presence of an antioxidant (N-acetyl cystein or NAC) or overexpression of superoxide dismutase (SOD1) [33]. Thus, in the yeast *Saccharomyces cerevisiae*, mitochondrial ROS are able to down-regulate mitochondrial biogenesis.

### 3.4. The Role of Glutathione in ROS-Induced Down-Regulation of Mitochondrial Biogenesis

One of the best indicators of the cellular redox state is the glutathione [48]. Indeed, the glutathione has a high concentration in the cytoplasm (of the order of 1mM) and the GSSG/GSH couple has a very high redox potential. Therefore, glutathione is considered one of the main cellular redox buffer [48]. The glutathione redox state is mostly maintained within the cell thanks to the glutathione reductase (GLR1), which reduces back the oxidized form (GSSG). Consequently, the glutathione redox state is decreased, i.e., more oxidized, in the null mutant for this enzyme. Even though this mutant is sensitive to oxidative stress [49], it is perfectly viable and the glutathione redox state can be increased in the mutant by addition of exogenous reduced glutathione to the culture medium [50]. We were able to show that the amount of transcription co-activator Hap4p (and consequently mitochondrial biogenesis) was decreased in the Δglr1 strain and its amount increased back when reduced glutathione was added to the culture medium, whereas ascorbate and lipoate had no effect on Hap4p amount either in the wild type cells or in the Δglr1 cells [32]. This supports a specific role of the glutathione redox state in the control of the biogenesis of the mitochondrial compartment through Hap4p.

## 4. ROS and Its Regulation of Mitochondrial Biogenesis in Mammalian Cells

### 4.1. Cellular cAMP Signaling and Mitochondrial Biogenesis

In mammalian cells, the consequences of the cAMP/PKA pathway signaling at the mitochondrial level has been very much studied and led to the discovery of a PKA and sAC (soluble adenylate cyclase) in the mitochondrial matrix [51,52] (Figure 6). It has been shown that induction of this mitochondrial matrix signaling pathway led to an increase of OXPHOS activity [53,54]. Intramitochondrial cAMP has been shown to regulate cytochrome c oxidase (complex IV). Indeed, cAMP/PKA-induced phosphorylation of complex IV subunit IV induces an increase in complex IV activity [55,56]. Moreover, PKA-dependent phosphorylation of several subunits of complex I appears to be involved in the assembly and enzymatic activity of this complex [57].

Beyond sAC and mitochondrial cAMP production, the cAMP/PKA pathway signaling regulates mitochondrial responses to cellular stimulations through cytosolic PKA. Upon signaling, PKA is mostly localized within membranes: both plasma and organelles membranes; of interest here is its localization in the mitochondrial membrane. This membranal localization requires an interaction with A-kinase anchor proteins (AKAP) [58,59]. In mitochondria, AKAPs are present in the outer membrane [60]. These anchor proteins tether PKA near cAMP production sites and PKA targets. This enhances cAMP transduction signal in apoptotic inhibition [61] and mitochondrial dynamics mechanisms [62] for example. Moreover, it has been shown that the cAMP/PKA pathway via the AKAPs can regulate the OXPHOS activity. PKA can phosphorylate a subunit of complex I, NDUFS4, which triggers its mitochondrial localization and a functional assembly of complex I [62,63]. On the other hand, in the presence of a high ATP/ADP ratio, PKA phosphorylates COX and promotes its inhibition by ATP. This allows a decrease in membrane potential and consequently in ROS production [64].

At the nuclear level, PKA phosphorylates and consequently activates cAMP response element binding protein (CREB), a transcriptional co-factor whose binding sequence is present on several promotors (Figure 6). Once activated, it initiates transcriptional cascades involved in multiple mechanisms such as mitochondrial biogenesis [65,66]. Indeed, the PGC-1α promotor as well as some mitochondrial genes, such as ND2, ND4, and ND5, encoding complex I subunits, have a CREB binding sequence [65]. Hence, activation of the cAMP pathway not only leads to mitochondrial biogenesis but also increases mitochondrial activities that are regulated by PKA induced phosphorylation [67]. An alteration of this pathway induces alterations of both mitochondrial respiration and ATP synthesis and enhancement of ROS production, that can be restored with restoration of the pathway [68].

It should be stressed here that CREB is a nuclear transcription factor that is able to localize in the mitochondria [69]. Indeed, despite the absence of mitochondrial localization signals on CREB, this transcription factor gets imported into mitochondria through the TOM complex in a membrane potential-dependent manner with the aid of mitochondrial matrix-residing heat shock protein, mtHSP70 [65]. Moreover, chromatin immunoprecipitation (ChIP) assay have evidenced CREB-binding sites on the D-loop of mitochondrial genome [70]. In order to decipher an eventual role of mitochondrial imported CREB, this transcription factor was depleted within the mitochondria. Such a depletion induced a decrease in the expression of several mitochondria-encoded RNAs of complex I with a concomitant reduction in complex I activity [65].

Mitochondrial dynamics (fission/fusion) and mtDNA maintenance are also essential for mitochondrial biogenesis. It has been shown that the cAMP/PKA pathway regulates mitochondrial dynamics thanks to PKA-induced DRP1 phosphorylation [71]. However, to date, very few data are available on the role of cAMP/PKA signaling on the molecular players of mitochondrial fusion. PKA can phosphorylate Mfn2, leading to cell growth arrest in rat vascular smooth muscle cells [72]. The relationships between cAMP signaling and mitochondrial dynamics have been reviewed in [73].

### 4.2. cAMP Signaling and Oxidative Stress

As stipulated above, the activities of at least two complexes of the respiratory chain have been shown to be regulated by cAMP-induced phosphorylation [57]. Such a regulation has consequences on mitochondrial ROS production that are highly dependent on both mitochondrial respiratory chain activity and the proton motive force across the mitochondrial membrane. Papa and his team showed that mitochondrial ROS production is enhanced under serum-limitation conditions in human and murine cells. This can be limited by addition of dibutyryl cAMP to the cells, a permeant derivative of cAMP that is able to activate PKA [74]. Moreover, in these conditions, cAMP has been shown to stimulate complex I activity and consequently mitochondrial respiratory chain activity [75]. Further studies have shown that in cancer cells (K-ras-transformed mouse fibroblast cells as well as in breast cancer cells), the cAMP/PKA pathway is altered [68]. This alteration is associated to an increase in ROS production and an alteration of mitochondrial activity. Moreover, the team of Papa showed that, in these different cell lines, Forskolin addition, an activator of AC, induces an increase in cellular cAMP level. This triggers an activation of the cAMP/PKA signaling pathway and consequently an increase in CREB phosphorylation and an enhancement in mitochondrial activity associated to a decrease in mitochondrial ROS [68]. Hence, cAMP pathway activation led to a decrease in mitochondrial ROS production.

These results clearly show that, similarly to what has been shown in the yeast *Saccharomyces cerevisiae*, a defect in the cAMP/PKA-induced phosphorylation of complexes of the respiratory chain leads to an increase in the oxidative stress at the mitochondrial level.

However, conflicting results regarding mitochondrial ROS production and the cAMP/PKA signaling pathway have been published. Indeed, a study by Westerblad and his team has shown that application of forskolin increases mitochondrial ROS production in isolated cardiomyocytes. Moreover, this increase in mitochondrial ROS can be prevented by PKA inhibitor pre-treatment [76], clearly showing it requires PKA activity. This result was further confirmed by Zhang et al. who showed that after an isoproterenol stimulation (that induces cAMP/PKA signaling) or forskolin treatment, mitochondrial ROS production is enhanced. In their study, the cAMP/PKA pathway triggered a rapid rise in mitochondrial ROS in neonatal murine cardiomycytes [77].

In conclusion, unlike what was shown in the yeast *Saccharomyces cerevisiae*, in mammalian cells, the role of cAMP/PKA pathway is much more ambiguous regarding the regulation of mitochondrial ROS production.

### 4.3. Oxidative Stress and Mitochondrial Biogenesis Regulation

In mammalian cells, it has been shown that mitochondrial biogenesis is tightly linked to ROS production [78,79,80]. In their studies, Lee et al. bring to light that an indoxyl sulfate-induced oxidative stress triggers a decrease in cellular mitochondrial content through a decrease in the biogenesis of this compartment in human umbilical vein endothelial cells. This ROS-induced trigger can be counteracted by antioxidant treatment such as NAC [81]. Similarly, resveratrol, a natural antioxidant, protects cells against HFD-induced apoptosis by reducing oxidative stress and stimulates mitochondrial biogenesis in T-reg [82].

Nevertheless, in neural cells, Spiegelman and colleagues show that upon oxidative stress expression of PGC-1α is enhanced as well as the expression of components of the mitochondrial ROS defense system such as SOD1, SOD2, catalase or glutathione peroxidase (GPX) [80]. Similar results have been shown in melanoma tumors. Indeed, in PGC-1α negative cells, ROS production is enhanced and induces apoptosis [83]. Moreover, Sharma et al. show, after a pre-treatment of rat brain cells with quercetin, a reduction of ROS production associated to an enhancement of PGC-1α expression and consequently an increase mitochondrial biogenesis [84].

Moreover, conflicting results were published: treatment of human preadipocytes with forskolin, which leads to an overactivation of the cAMP/PKA pathway -and a decrease in mitochondrial ROS production-, increased mitochondrial DNA copy number [85]. Further, in HeLa cells, respiratory uncoupling, which is well known to decrease mitochondrial reactive oxygen species production, activates NRF-1 (nuclear respiratory factor-1) [86].

Hence, in mammalian cells, ROS induced regulation of mitochondrial biogenesis can lead to either an increase or a decrease of this process in response to an increase of mitochondrial ROS.

One possible explanation for such conflicting results in the literature might be that the effect of oxidative stress on mitochondrial biogenesis depends on the severity and duration of the stress. Acute/mild stress can stimulate PGC-1α expression and mitochondrial biogenesis due to the involvement of mitochondrial quality control [80]. However, severe and permanent stress can have the opposite effects. For example, heart failure induced by aortic banding [87] and myocardial infarction [88] decreased mitochondrial transcription factors, including PGC-1α as well as NRF1, NRF2, MTF. This decrease in mitochondrial transcription factors is associated with a decrease in mitochondrial content/activities.

## 5. Conclusions

It is well known mitochondria are the first site of ROS production. The generation of these species is controlled by multiple parameters. In mammalian cells as well as in yeast, ROS-induced signaling is able to regulate mitochondrial biogenesis. In yeast, the role of ROS is primarily a down-regulation by its action on Hap4p, the main regulator of mitochondrial biogenesis (homologous of PGC-1α). In mammalian, the situation is less straightforward: ROS formation can trigger PGC-1α induction and induces an enhancement of mitochondrial biogenesis, however PGC-1α is also induced under uncoupling conditions where mitochondrial ROS production rate are the lowest. Mitochondrial ROS are clearly involved in the cross talk between this compartment and the nucleus. However, conflicting results regarding their role in the regulation of mitochondrial biogenesis in mammalian cells show that further studies will be necessary to clarify the situation.

## Figures and Tables

**Figure 1 cells-08-00287-f001:**
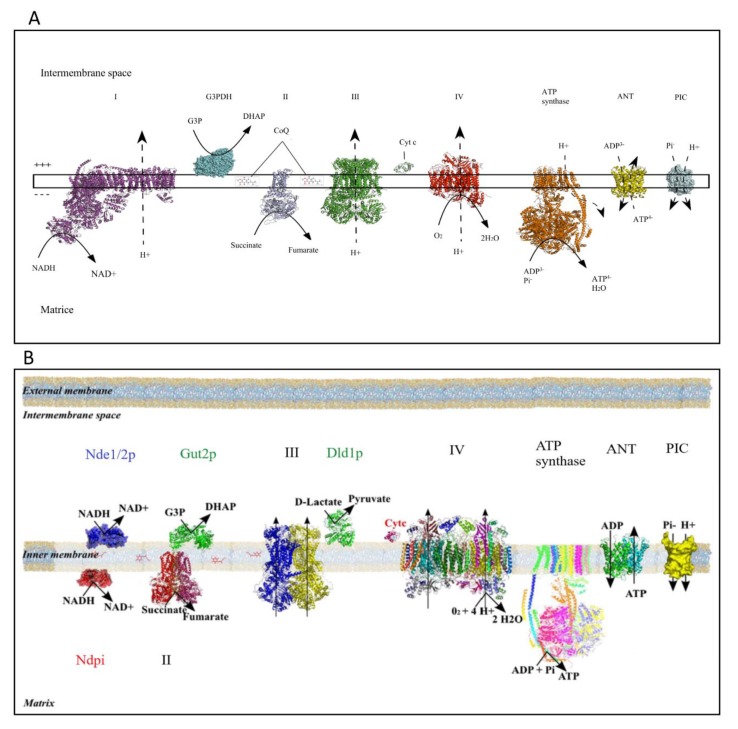
The oxidative phosphorylation system (OXPHOS) system in mammalian cells (**A**) and in yeast (**B**) (modified from Rigoulet et al. 2010 [2]). Numbers represent respiratory chain complexes. I: proton-pumping nicotinamide adenine dinucleotide (NADH) dehydrogenase, II: Succinate dehydrogenase; III: Cytochrome c reductase; IV: Cytochrome c oxidase. Adenine nucleotide translocator (ANT) and Phosphate inorganic carrier are mitochondrial carriers. Gut2p is the glycerol-3-phosphate dehydrogenase in yeast. Dld1p is the lactate dehydrogenase in yeast.

**Figure 2 cells-08-00287-f002:**
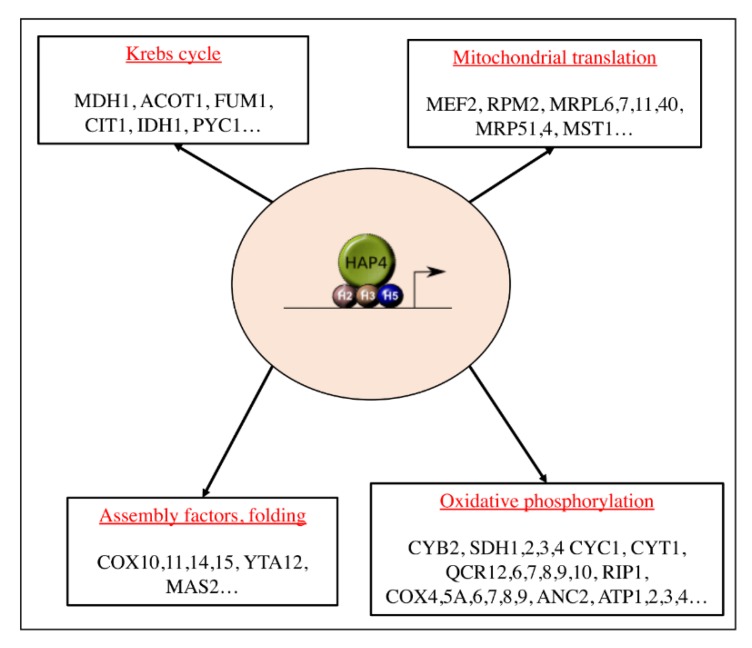
The heme activator proteins (HAP) complex. The master regulator of mitochondrial biogenesis in the yeast *Saccharomyces cerevisiae*. The four subunits constituting the complex are represented here, HAP2 (H2), HAP3 (H3), and HAP5 (H5) are the DNA-binding subunits and HAP4 is the activating subunit. Modified from [24].

**Figure 3 cells-08-00287-f003:**
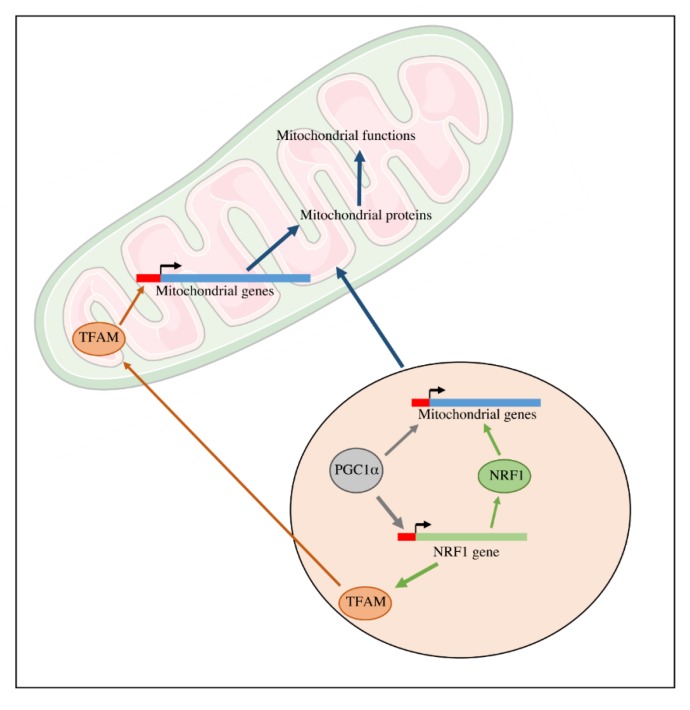
Transcriptional regulation of mitochondrial biogenesis. The expression of mitochondrial genes encoded by the nuclear genome is regulated by transcriptional factors such as NRF1 and its coactivator PGC1α. TFAM is implicated in the expression of the genes encoded by the mtDNA.

**Figure 4 cells-08-00287-f004:**
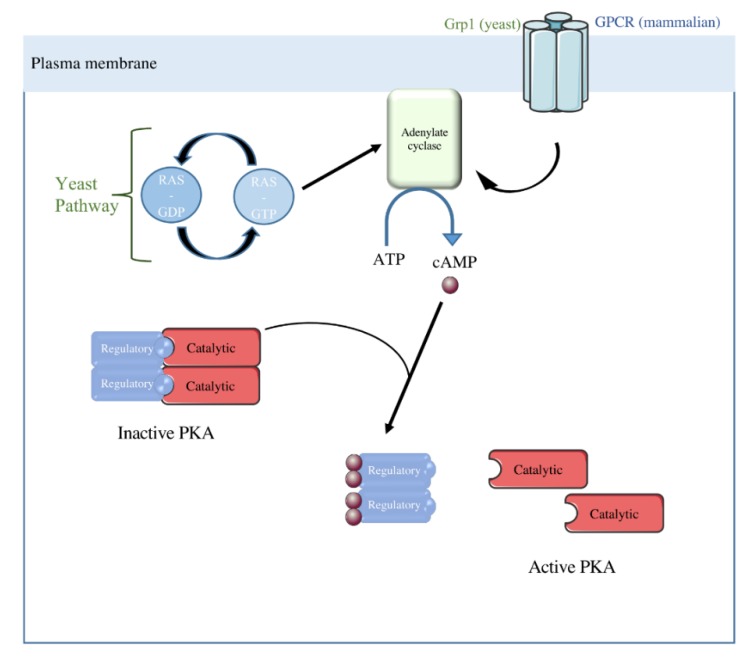
The Ras-cAMP/PKA Pathway in yeast and mammalian cells. In yeast, adenylate cyclase can be activated by two systems: Grp1 or the RAS pathway. In mammalian cells, adenylate cyclase activity is controlled by GPCR. cAMP synthesized by the adenylate cyclase binds to PKA regulatory subunits leading to the activation of the PKA catalytic subunit thanks to its dissociation from its regulatory subunits.

**Figure 5 cells-08-00287-f005:**
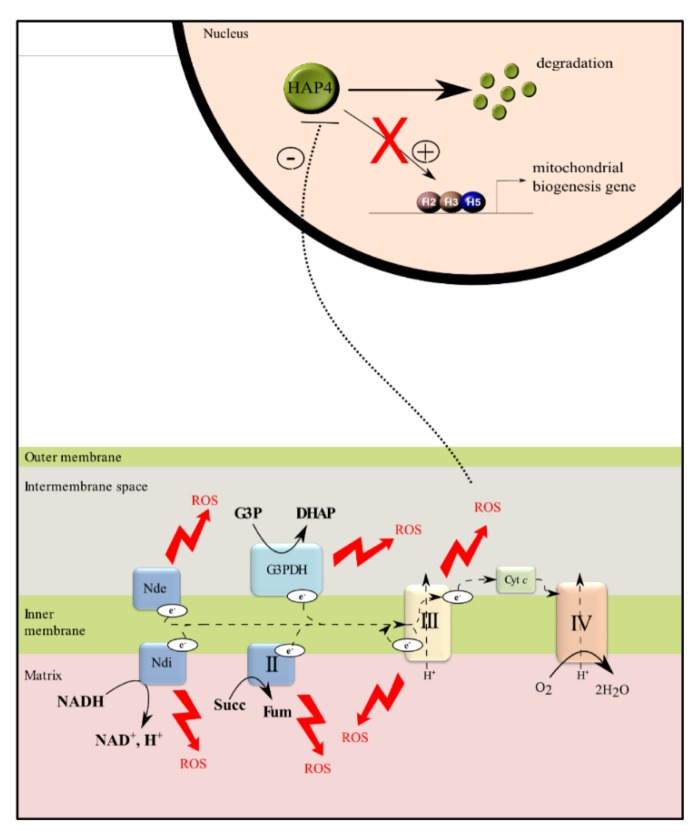
Regulation of mitochondrial biogenesis by mitochondrial reactive oxygen species (ROS) in *Saccharomyces cerevisiae*. All dehydrogenases are able to generate ROS. Mitochondrial ROS production induces an enhancement of HAP4 degradation and consequently a decrease in mitochondrial biogenesis.

**Figure 6 cells-08-00287-f006:**
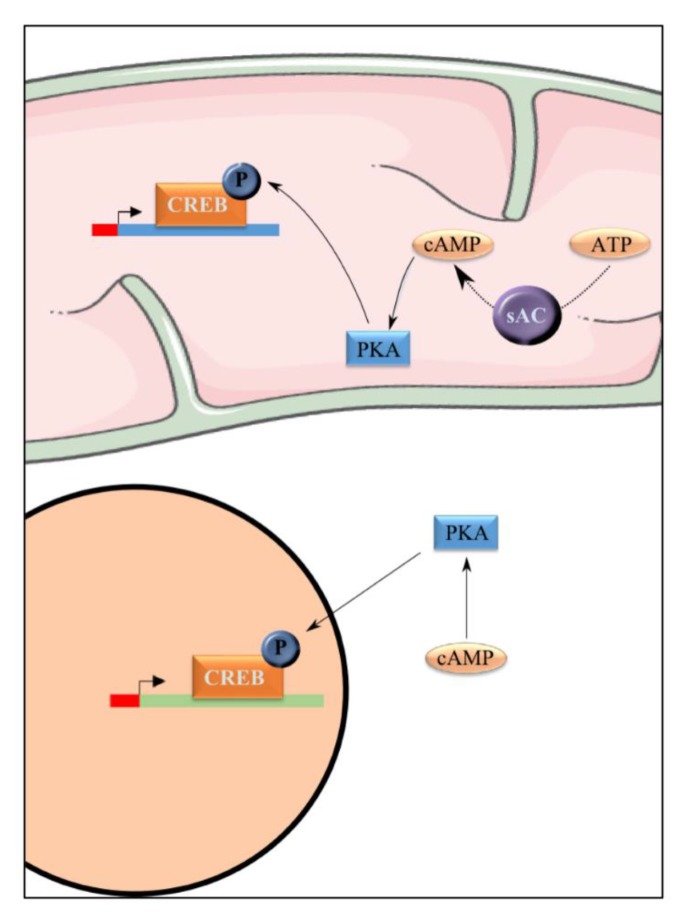
PKA actives CREB. PKA is able to phosphorylate cAMP response element binding protein (CREB) and consequently activates this transcription co-factor. CREB is present in the nucleus and in the mitochondria.

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
