# Peer review of "Mitochondrial Biogenesis and Mitochondrial Reactive Oxygen Species (ROS): A Complex Relationship Regulated by the cAMP/PKA Signaling Pathway"

_cells, 2019, doi:10.3390/cells8040287_

Round 1

Reviewer 1 Report

Fig. 1 and 2 are very similar to those of  Ref. [31],

Yoboue, E.D. ; Devin, A. Reactive oxygen species-mediated control of mitochondrial biogenesis. Int. J. 381 Cell Biol. 2012, 2012

If the authors cite this paper in Fig. 1 and 2 Legends, like "reproduced from Ref. [31]", then everythng is fine. And it will be no more problem, and there will be no self-plagiarism.

Author Response

This is now done

Reviewer 2 Report

The manuscript review by Bouchez et al.  describes the role of cAMP/PKA pathway in the regulation of key cell functions related to oxidative metabolism, stress response and mitochondrial biogenesis. This is an interesting review article which includes relevant published work and molecular models of the cAMP-mediated regulation of metabolism, both in yeast as well as in mammalian systems with potential relevance to a wide audience. 

The manuscript review is well written and clear. 

The major point that needs to be included in the present manuscript is the relevance of mammalian mitochondria-anchored cAMP scaffold proteins (AKAPs) in the control of organelle activities. Beyond sAC and intramitochondrial cAMP production, work from different groups have mechanistically addressed the biological relevance of AKAPs/PKA complexes anchored at the outer mitochondrial membrane in key signaling pathways leading to organelle biogenesis, stress response, mitochondrial dynamics and oxidative metabolism. This is an important issue that cannot be ignored when addressing the cAMP action at the mitochondrial compartment. A long list of published papers in this matter is currently available.

Author Response

The major point that needs to be included in the present manuscript is the relevance of mammalian mitochondria-anchored cAMP scaffold proteins (AKAPs) in the control of organelle activities. Beyond sAC and intramitochondrial cAMP production, work from different groups have mechanistically addressed the biological relevance of AKAPs/PKA complexes anchored at the outer mitochondrial membrane in key signaling pathways leading to organelle biogenesis, stress response, mitochondrial dynamics and oxidative metabolism. This is an important issue that cannot be ignored when addressing the cAMP action at the mitochondrial compartment. A long list of published papers in this matter is currently available.

This issue has been taken care off and the following text has been added to our review:

Beyond sAC and mitochondrial cAMP production, the cAMP/PKA pathway signaling regulates mitochondrial responses to cellular stimulations through cytosolic PKA. Upon signaling PKA is mostly localized within membranes: both plasma and organelles membranes, of interest here is it localization in the mitochondrial membrane. This membranal localization requires an interaction with A-kinase anchor proteins (AKAP) [57], [58]. In mitochondria, AKAPs are present in the outer membrane [59]. These anchor proteins tether PKA near cAMP production sites and PKA targets. This enhances cAMP transduction signal in apoptotic inhibition [60] and mitochondrial dynamics mechanisms [61] for example. Moreover, it has been shown that the cAMP/PKA pathway via the AKAPs can regulate the OXPHOS activity. PKA can phosphorylate a subunit of complex I, NDUFS4, which triggers its mitochondrial localization and a functional assembly of complex I [61], [62]. On the other hand, in the presence of a high ATP/ADP ratio, PKA phosphorylates COX and promotes its inhibition by ATP. This allows a decrease in membrane potential and consequently in ROS production [63].

Reviewer 3 Report

The MS is written well and contain a discussion of previous studies on the role of the cAMP-PKA pathway in mitochondrial biogenesis-ROS crosstalk. We have several comments to be taken into consideration before the MS is recommended for publication.

Major comments:

1. Authors discuss the role of ROS in the regulation of mitochondrial biogenesis mostly in yeast (line 118 to 216) and only briefly focus on mammalian cells (lines 217 to 299). Besides, the authors mainly discuss their studies. Discussion of others’ studies could improve the significance and readiness of the MS.

2. Statements “All 13mRNAs encode subunits of the OXPHOS” (line 77) and “The mtDNA encodes 13 components of OXPHOS system complexes” (line 109) are not correct since 13 mRNAs of mtDNA encode 11 subunits of the ETC complexes I (7), III (1) and IV (3), and 2 subunits of ATP synthase (complex V), which is responsible for OXPHOS. ETC activity occurs at complexes I to IV, OXPHOS at complex V. Likewise, statement  

3. Authors discuss a possible role of PKA to phosphorylate transcription factors involved in mitochondrial biogenesis (lines 231-240). It is not clear whether PGC-1a (or PPARa), a key regulator of mitochondrial biogenesis, is phosphorylated (since PKA is important for phosphorylation other PTMs can be ignored). For example, H2O2-induced oxidative stress in H9c2 cardioblasts stimulated phosphorylation of PGC-1alpha and PPARa (PMID: 25617357). Moreover, this study demonstrated translocation of PPARa to the mitochondria under oxidative stress and its interaction with cyclophilin D, a major regulator of the permeability transition pore. Direct activation of PGC-1a through its phosphorylation by AMPK, a downstream kinase of PKA (PMID: 22553202), was shown in skeletal muscle (PMID: 17609368). These studies among others should be discussed in one paragraph.

3. Authors indicate the complexity of the effect of oxidative stress on the expression of mitochondrial transcription factors, particularly, PGC-1a. Indeed, the effect of oxidative stress on mitochondrial biogenesis depends on the severity and duration of the stress. Acute/mild stress can stimulate PGCa expression and mitochondrial biogenesis due to the involvement of mitochondrial quality control. However, severe and permanent stress can have the opposite effects. For example, heart failure induced by aortic banding (PMID: 12824444) and myocardial infarction (PMID: 16679399) decreased mitochondrial transcription factors, including PGC-1alpha as well as NRF1, NRF2, MTF.

Minor comments:

1. Lines 105-106: Revise “Moreover, not only do PGC-1 transcription factors activators interact with transcription factors such as NRF1 and 2 but they…”.

2. Line 90: Change “These factors allow protein…” to “These factors regulate protein…”

3. Line 78: Change “…genetically semiautonomous in that they rely strongly…” to “…genetically semiautonomous and rely strongly…”

4. Line 45: Change “…adenine nucleotide carrier (ANC)” to “adenine nucleotide translocator (ANT)”

5. The quality of all Figures should be improved by using the same style and fonts.

6. All abbreviation should be mentioned at the first meeting in the text after the full name. E.g., there is no full name of PGC-1a at first meeting (line 85).

Author Response

1.     Authors discuss the role of ROS in the regulation of mitochondrial biogenesis mostly in yeast (line 118 to 216) and only briefly focus on mammalian cells (lines 217 to 299). Besides, the authors mainly discuss their studies. Discussion of others’ studies could improve the significance and readiness of the MS.

This issue has been taken care off and the following text has been added to our review:

In mammalian cells, it has been shown that mitochondrial biogenesis is tightly linked to ROS production [77]–[79]. In their studies, Lee et al. bring to light that an indoxyl sulfate-induced oxidative stress triggers a decrease in cellular mitochondrial content through a decrease in the biogenesis of this compartment in human umbilical vein endothelial cells. This ROS-induced trigger can be counteracted by antioxidant treatment such as NAC [80]. Similarly, resveratrol, a natural antioxidant, protects cells against HFD-induced apoptosis by reducing oxidative stress and stimulates mitochondrial biogenesis in T-reg [81].

 Nevertheless, in neural cells, Spiegelman and colleagues show that upon oxidative stress expression of PGC1α is enhanced as well as the expression of components of the mitochondrial ROS defense system such as SOD1, SOD2, catalase or glutathione peroxidase (GPX) [79]. Similar results have been shown in melanoma tumors. Indeed, in PGC1α negative cells, ROS production is enhanced and induces apoptosis [82]. Moreover, Sharma et al. show, after a pre-treatment of rat brain cells with quercetin, a reduction of ROS production associated to an enhancement of PGC1α expression and consequently an increase mitochondrial biogenesis [83].

Moreover, conflicting results were published: treatment of human preadipocytes with forskolin, which leads to an overactivation of the cAMP/PKA pathway-and a decrease in mitochondrial ROS production-, increased mitochondrial DNA copy number [84]. Further, in HeLa cells, respiratory uncoupling, which is well known to decrease mitochondrial reactive oxygen species production, activates NRF-1 (nuclear respiratory factor-1) [85].

Hence, in mammalian cells, ROS induced regulation of mitochondrial biogenesis can lead to either an increase or a decrease of this process in response to an increase of mitochondrial ROS.

2.   Statements “All 13mRNAs encode subunits of the OXPHOS” (line 77) and “The mtDNA encodes 13 components of OXPHOS system complexes” (line 109) are not correct since 13 mRNAs of mtDNA encode 11 subunits of the ETC complexes I (7), III (1) and IV (3), and 2 subunits of ATP synthase (complex V), which is responsible for OXPHOS. ETC activity occurs at complexes I to IV, OXPHOS at complex V. Likewise, statement  

This has been corrected

3. Authors discuss a possible role of PKA to phosphorylate transcription factors involved in mitochondrial biogenesis (lines 231-240). It is not clear whether PGC-1a (or PPARa), a key regulator of mitochondrial biogenesis, is phosphorylated (since PKA is important for phosphorylation other PTMs can be ignored). For example, H2O2-induced oxidative stress in H9c2 cardioblasts stimulated phosphorylation of PGC-1alpha and PPARa (PMID: 25617357). Moreover, this study demonstrated translocation of PPARa to the mitochondria under oxidative stress and its interaction with cyclophilin D, a major regulator of the permeability transition pore. Direct activation of PGC-1a through its phosphorylation by AMPK, a downstream kinase of PKA (PMID: 22553202), was shown in skeletal muscle (PMID: 17609368). These studies among others should be discussed in one paragraph.

We thank the reviewer for pointing this out. However, we tried in our review to take into consideration papers where the regulation of mitochondrial biogenesis was actually shown.  For example, in PMID: 17609368 what was studied was the post-translational modifications of PGC1a in response to AMPK activation. No cAMP signaling and/or actual regulation of mitochondrial biogenesis is studied in this paper. We did not in this review approach the question of PGC1a post-translational modifications and their role in the regulation of mitochondrial biogenesis.

3.     Authors indicate the complexity of the effect of oxidative stress on the expression of mitochondrial transcription factors, particularly, PGC-1a. Indeed, the effect of oxidative stress on mitochondrial biogenesis depends on the severity and duration of the stress. Acute/mild stress can stimulate PGCa expression and mitochondrial biogenesis due to the involvement of mitochondrial quality control. However, severe and permanent stress can have the opposite effects. For example, heart failure induced by aortic banding (PMID: 12824444) and myocardial infarction (PMID: 16679399) decreased mitochondrial transcription factors, including PGC-1alpha as well as NRF1, NRF2, MTF.

We would like to thank the reviewer for this very pertinent remark. This remark has been considered and added to our review as well as the references suggested by this reviewer.

Minor comments:

1. Lines 105-106: Revise “Moreover, not only do PGC-1 transcription factors activators interact with transcription factors such as NRF1 and 2 but they…”.

This is now done

2. Line 90: Change “These factors allow protein…” to “These factors regulate protein…”

            This is now done

3. Line 78: Change “…genetically semiautonomous in that they rely strongly…” to “…genetically semiautonomous and rely strongly…”

            This is now done

4. Line 45: Change “…adenine nucleotide carrier (ANC)” to “adenine nucleotide translocator (ANT)”

This is now done

5. The quality of all Figures should be improved by using the same style and fonts.

This is now done

6. All abbreviation should be mentioned at the first meeting in the text after the full name. E.g., there is no full name of PGC-1a at first meeting (line 85).

This is now done

Round 2

Reviewer 2 Report

no further comments

This manuscript is a resubmission of an earlier submission. The following is a list of the peer review reports and author responses from that submission.

Round 1

Reviewer 1 Report

Review Article

Yoboue ED, Devin A.

Reactive oxygen species-mediated control of mitochondrial biogenesis.

Int J Cell Biol. 2012;2012:403870. doi: 10.1155/2012/403870. Epub 2012 May 30.

PMID:22693510

This article contains similar content and discussion.

Author Response

We would like to thank the reviewer for his review and his comment. We do agree that part of our review might appear a bit redondant with one of our previous review. However in this specific manuscript we analyse the relationships between ROS and mitochondrial biogenesis but more importantly, as specified in the title, the role of the cAMP/PKA pathway in this process, which has not been reviewed previously.

Reviewer 2 Report

This is an interesting review on the interplay between cAMP/PKA and ROS, and their influences on mitochondrial biogenesis in yeast and mammalian cells. The review is a necessary complement to current literature in the field that predominantly focuses on mammalian systems.  Overall, the manuscripts is nicely composed. Nevertheless, I hope the authors could address a few points below.

1.    The authors state that “The main superoxide (O.-2) producer in the cell is the mitochondrial respiratory chain” (page 2, line 55). Accordingly, one shall expect that less mitochondrial would produce less ROS. Then, why Tpk3p mutant cells that have reduced mitochondrial content, have increased ROS production? Perhaps, there are more defects other than reduced mitochondrial content in Tpk3p mutant?  Could the imbalance between nuclear encoded and mitochondrial-encoded ETC components, or a lack of coordination between TCA cycle and ETC  contributes to the increased ROS production? Maybe the authors cold soften the statement that respiratory chain is the main producer of ROS? After all, other enzymes also produce ROS.

2.    In mammalian system, ROS is considered as a mitochondrial stress signal to stimulate mitochondria biogenesis. This review summarizes evidences in yeast painting a different picture--a feed-forward loop between ROS and mitochondrial biogenesis. Impaired mitochondrial biogenesis increases ROS production, which in turn inhibits HAP4 activity that stimulates mitochondrial biogenesis. What is the physiological significance of this regulation? I am not an expert on yeast metabolism. It would be interesting if the authors could provide their take on this aspect. 

3.    Mitochondrial dynamics (fission/fusion) and mtDNA maintenance are also essential for mitochondrial biogenesis and have been linked to cAMP/PKA signaling. Although they are not the focus of this review, it seems more complete if authors could touch on these topics little bit.

Minor points:

1.    Page 1, line 36: the name of FADH2 should be in English and be consistent with the Abbreviation section.

2.    Page 3, line 78: it would be better if authors describe mtDNA gene contents here, which would help general audience to understand the meaning of mito-nuclear coordination.

3.    Page 3, line 79:  there is a repeated phrase of “numerous mitochondrial proteins”.

4.    It is strange to have Figure 2A and 2B on different pages. They should appear as Figure 2 and 3. 

5.    Page 5, line 135 and 136: “regulator” should be “regulatory”, also to be consistent with the term used in the figure legend.

Author Response

This is an interesting review on the interplay between cAMP/PKA and ROS, and their influences on mitochondrial biogenesis in yeast and mammalian cells. The review is a necessary complement to current literature in the field that predominantly focuses on mammalian systems.  Overall, the manuscripts is nicely composed. Nevertheless, I hope the authors could address a few points below.

1.    The authors state that “The main superoxide (O.-2) producer in the cell is the mitochondrial respiratory chain” (page 2, line 55). Accordingly, one shall expect that less mitochondrial would produce less ROS. Then, why Tpk3p mutant cells that have reduced mitochondrial content, have increased ROS production? Perhaps, there are more defects other than reduced mitochondrial content in Tpk3p mutant?  Could the imbalance between nuclear encoded and mitochondrial-encoded ETC components, or a lack of coordination between TCA cycle and ETC  contributes to the increased ROS production? Maybe the authors cold soften the statement that respiratory chain is the main producer of ROS? After all, other enzymes also produce ROS.

 We would like to thank the reviewer for this very pertinent comment. We have modified the text in order for it to be more comprehensive. Actually the priming event is the increase in mitochondrial ROS due to a defect in Tpk3p-induced phosphorylation at the level of mitochondrial respiratory chain. This increase then leads to a decrease in mitochondrial biogenesis. This was showed in reference 43 and the above sentence has been added to the text. We do agree that less mitochondria should produce less ROS. However the increase in ROS production in the ∆Tpk3p mitochondria is such that it is maximum (equivallent to Antimycin A condition) in these mitochondria and consequently higher than in the wild type mitochondria. We do agree that other enzymes produce ROS however, in our hands experiments in these mutants mitochondria could only pinpoint an increase in respiratory chain-produced ROS. Last, our analysis of mitochondria from the delta Tpk3p mutant showed that, at least in terms of mitochondrial cytochromes, these mitochondria’s respiratory chain are not defective i.e. the cells have less mitochondria but the mitochondrial content in cytochromes is comparable to the wild type (43).

2.    In mammalian system, ROS is considered as a mitochondrial stress signal to stimulate mitochondria biogenesis. This review summarizes evidences in yeast painting a different picture--a feed-forward loop between ROS and mitochondrial biogenesis. Impaired mitochondrial biogenesis increases ROS production, which in turn inhibits HAP4 activity that stimulates mitochondrial biogenesis. What is the physiological significance of this regulation? I am not an expert on yeast metabolism. It would be interesting if the authors could provide their take on this aspect. 

  We thank this reviewer for pointing this major difference between the two models out. First as pointed in our answer to his first question in yeast, the increase in mitochondrial ROS production is the priming event that induces a decrease in mitochondrial biogenesis (this is shown in the ∆tpk3 cells but also by increasing mitochondrial ROS production with inhibitors in the wild type cells ref 44). Second, this excellent question made us realize that the picture we painted regarding cAMP signaling and oxidative stress was incomplete in that in mammalian cells both an increase or a decrease in mitochondrial ROS production can be assessed upon cAMP signaling. This led us to complete this section of our revue with a few references. To answer this reviewer’s question, it seems that a unicellular eukaryote will try to get rid of “unfit” mitochondria given that it can slow down its growth (decrease in ATP demand). This might not be true for a more complex organism that requires a constant ATP supply and where a mitochondrial deficiency (ROS/Respiratory rate/ATP synthesis) is generally “compensated” by an increase in mitochondrial amount.

3.    Mitochondrial dynamics (fission/fusion) and mtDNA maintenance are also essential for mitochondrial biogenesis and have been linked to cAMP/PKA signaling. Although they are not the focus of this review, it seems more complete if authors could touch on these topics little bit.

Very few data are available on mitochondrial dynamics and cAMP signaling. To our knowledge this has never been shown in yeast. In mammalian cells, DRP1 has been shown to be phosphorylated through cAMP signaling, as well as Mfn2. The physiological relevance of these phosphorylations remains to be investigated. We did add a few sentences on this subject in our manuscript as well as a couple of references.

Reviewer 3 Report

The review manuscript by Bouchez and  Devin   summarize our current knowledge on the mitochondrial biogenesis, mitochondrial ROS and cAMP signaling. These topics are very interesting but they are all covered by this manuscript in superficial manner.  

The manuscript is articulate in sections regarding Saccharomyces and section regarding mammalian cells, probably rewritting the manuscript  on only one species  it will be possible to deepen the topics better, in order to  make the manuscript more interesting for the reader.

Author Response

            We thank the reviewer for his remark. However one of the goals of this manuscript was indeed to compare yeast and mammalian cells and to show the relevance of yeast as a model system. Thus, taking out the mammalian cells part would alter our message. Moreover, for the parts that may be deemed as superficial, we took great care to use the proper references within the text and hope the reader will refer to these references for more in depth analysis of the process under consideration.

Reviewer 4 Report

This review summarized the relationship between mitochondrial biogenesis and mitochondrial ROS and the regulation by cAMP/PKA signaling in both yeasts and mammal cells.  This review is of major importance for mitochondrial study.

The following are several major concerns.

1 The authors have published a review about mitochondrial biogenesis and ROS in yeasts in 2014 (Yoboue ED et al. Biochimica et Biophysica Acta 1837 (2014) PMID: 24602596). It would be better to add more advancements in this field especially after 2014.  

2 In page 10, it only introduced the finding that cAMP/PKA signaling pathway decrease mitochondrial ROS.  However, cAMP/PKA signaling has also been shown to induce mitochondrial ROS production in some mammal cells (Zhang J et al. Biochem Biophys Res Commun. 2017, PMID: 28552530).

3 Considering the diversity of signaling and mitochondrion function in different mammal cell-types and different diseases. It is recommended to clearly indicate the cell-type and disease model when summarizing the studies on mammal cells.

Author Response

1 The authors have published a review about mitochondrial biogenesis and ROS in yeasts in 2014 (Yoboue ED et al. Biochimica et Biophysica Acta 1837 (2014) PMID: 24602596). It would be better to add more advancements in this field especially after 2014.  

            We do agree with this reviewer. However, careful analysis of the litterature since 2014 shows that there were no major advancements in this field since 2014 and our manuscript on this subject is currently under review. Further, the current review aims not only to assess the state of the art on mitochondrial biogenesis and ROS but also the pre-eminent role of cAMP signaling-induced regulation is this process and the relevance of the yeast S. cerevisiae as a model system.

2 In page 10, it only introduced the finding that cAMP/PKA signaling pathway decrease mitochondrial ROS.  However, cAMP/PKA signaling has also been shown to induce mitochondrial ROS production in some mammal cells (Zhang J et al. Biochem Biophys Res Commun. 2017, PMID: 28552530).

We would like to thank this reviewer for pointing out a major oversight from us. This has now been added to the text in a paragraph within the section cAMP signaling and oxidative stress.

3 Considering the diversity of signaling and mitochondrion function in different mammal cell-types and different diseases. It is recommended to clearly indicate the cell-type and disease model when summarizing the studies on mammal cells.

This is a very pertinent remark and has now been done within the text

Round 2

Reviewer 1 Report

The authors already published the similar paper as attached, and even they did not cite this paper.

Reviewer 3 Report

My opinion is that the revised version of the manuscript is still lacking in informations.